# Structural Color of Partially Deacetylated Chitin Nanowhisker Film Inspired by Jewel Beetle

**DOI:** 10.3390/ma17215357

**Published:** 2024-11-01

**Authors:** Dagmawi Abebe Zewude, Masaaki Akamatsu, Shinsuke Ifuku

**Affiliations:** 1Organization for Research Initiative and Promotion, Tottori University, 4-101 Koyama-Minami, Tottori 680-8550, Japan; dagmawi.abe@tottori-u.ac.jp (D.A.Z.); makamatsu@tottori-u.ac.jp (M.A.); 2Center for Research on Green Sustainable Chemistry, Tottori University, Tottori 680-8550, Japan; 3Research Institute for Sustainable Humanosphere, Kyoto University, Uji 611-0011, Japan

**Keywords:** nano-chitin, nanofiber, nano-whisker, color structure, multilayer structure

## Abstract

Nanochitin was developed to effectively utilize crab shells, a food waste product, and there is ongoing research into its applications. Short nanowhiskers were produced by sonicating partially deacetylated nanochitin in water, resulting in a significant decrease in viscosity due to reduced entanglement of the nanowhiskers. These nanowhiskers self-assembled into a multilayered film through an evaporation technique. The macro- and nanoscale structures within the film manipulate light, producing vibrant and durable structural colors. The dried cast film exhibited green and purple stripes extending from the center to the edge formed by interference effects from the multilayer structure and thickness variations. Preserving structural colors requires maintaining a low ionic strength in the dispersion, as a higher ionic strength reduces electrostatic repulsion between nanofibers, increasing viscosity and potentially leading to the fading of color. This material’s sensitivity to environmental changes, combined with chitin’s biocompatibility, makes it well-suited for food sensors, wherein it can visually indicate freshness or spoilage. Furthermore, chitin’s stable and non-toxic properties offer a sustainable alternative to traditional dyes in cosmetics, delivering vivid and long-lasting color.

## 1. Introduction

Chitin is a natural carbohydrate polymer composed of N-acetylglucosamine repeating units with acetamide groups. It is commonly found as a structural component in the exoskeletons of arthropods, including crustaceans such as crabs and shrimp [1] and insects like beetles [2]. Although chitin is derived from various sources, crab and shrimp shells are the primary sources for commercial production [3]. Within the exoskeleton, chitin forms nanoscale fibers arranged in a hierarchical structure [4]. Mechanical treatment disrupts this assembly, isolating individual nanofibers [5], which can be readily dispersed in water. This high dispersibility facilitates the formation of chitin into various molded shapes, including microfibers [6], sponges, sheets, and films [7].

The versatility and excellent formability of chitin nanofibers make them suitable for a broad range of applications. Chitin nanofibers and their partially deacetylated derivatives have demonstrated several biological functions, such as wound healing [8], hair growth promotion [9], and anti-inflammatory effects [10]. They also show promise in the development of smart materials [11], highlighting chitin’s potential as a sustainable, multifunctional material.

The exoskeletons of certain insects, such as beetles and the blue morpho butterfly, contain uniquely organized chitin structures that produce iridescent colors through interactions with light and their ordered nanostructures [12]. In nature, nanoscale chitin fibers are arranged in precise patterns to manipulate light via interference, diffraction, and scattering, resulting in the reflection of specific wavelengths and the creation of vibrant colors [13]. One of the simplest examples of such structures is a thin film, which generates color through the interference of light reflecting off its top and bottom surfaces, as commonly observed in soap bubbles [14]. When multiple thin films with different refractive indices are layered together, they form a multilayer structure [13].

A notable example is the jewel beetle (*Sternocera aequisignata*), which exhibits intense iridescent colors due to the periodic helicoidal arrangement of chitin in its elytron, allowing it to interact with light in a way that produces a striking green hue [15,16]. This effect, known as multilayer interference, occurs when periodic structures within the chitin match the wavelengths of visible light, resulting in the beetle’s characteristic coloration, as shown in Figure 1 [17].

Drawing inspiration from natural phenomena, multilayered films can be created from chitin nanowhisker suspensions using an evaporation-assisted method. Studies have shown that chitin nanofibers can self-assemble into layered structures through evaporation [18]. However, maintaining precise control over the spacing between layers (pitch length) within the range of visible light remains a significant challenge [13]. By comparison, cellulose nanofibers can more readily form chiral nematic liquid crystals above a certain concentration threshold, leading to photonic films that exhibit vibrant colors as the dispersion evaporates [19,20]

To promote chitin self-assembly, hydrolysis with strong acids like HCl is commonly used [18]. This study, however, explores an alternative technique: deacetylation, which converts chitin’s acetamide groups to amino groups through treatment with concentrated NaOH. When placed in acidic water, the amino groups (-NH_2_) are protonated to (-NH_3_^+^), creating electrostatic repulsion that aids in dispersion. Ultrasonication further improves nanofiber mobility by minimizing entanglement, resulting in a more even dispersion [21]. This method supports the rapid formation of structurally colored films with periodic structures and pitch lengths suitable for visible light.

Colors derived from this structural approach provide a durable, eco-friendly, and non-toxic alternative to synthetic pigments, offering vibrant, customizable hues without requiring harmful chemicals [22]. Chitin nanofibers, known for their biocompatibility, biodegradability, and abundant availability, offer a sustainable choice for creating structural colors. Their responsiveness to environmental conditions makes them ideal for food sensors, providing clear visual indicators of freshness and spoilage. This environmentally friendly approach also serves as a safe substitute for conventional dyes in cosmetics, producing vivid, long-lasting, and non-toxic hues suitable for skin contact.

## 2. Materials and Methods

### 2.1. Materials

A partially deacetylated chitin nanofiber suspension in aqueous lactic acid solution (pH 4.0) was purchased from Marin Nanofiber Co., Ltd. (Tottori, Japan). The degree of deacetylation was 50%. Acetic acid and sodium chloride were purchased from FUJIFILM Wako Pure Chemical Corporation (Osaka, Japan) and Nacalai Tesque Inc. (Kyoto, Japan). Jewel beetle (*Sternocera aequisignata*) elytra were collected from Thailand.

### 2.2. Preparation of Self-Assembled Iridescent Film from Partially Deacetylated Chitin Nanowhiskers

A 30 g suspension of partially deacetylated chitin nanofibers, containing 1.4 wt% solids and possessing a pH of 3.86, was sonicated for 8 min at a 50% duty cycle and power level 6 using a Branson Sonifier 250 ultrasonic generator (Danbury, CT, USA). Aliquots of the sonicated dispersion (1.0, 3.0, and 4.0 g) were transferred into polystyrene Petri dishes (52.4 mm in diameter), agitated to achieve even dispersion, and then dried in an incubator at 60 °C. Once dried, the thin films were easily removed from the dish surface.

To examine the effect of ionic strength on the self-assembly of partially deacetylated chitin nanofibers, the 30 g suspension was subjected to an initial 8 min sonication, followed by dialysis in aqueous NaCl solutions at concentrations of 1, 6, 10, 20, and 44 mM and adjusted to a pH of 3. Each sample was then subjected to an additional 1 min sonication. Subsequently, 1 g of each suspension was placed in a polystyrene Petri dish, dried at 60 °C in an incubator, and peeled off for further characterization.

### 2.3. Characterization of Partially Deacetylated Chitin Nanowhiskers and Iridescent Film

Field-emission scanning electron microscopy (FE-SEM, JSM-6701F, JEOL, Tokyo, Japan) was used to observe the morphology of the partially deacetylated chitin nanofibers and measure the pitch length of the films. Imaging was carried out in vacuum mode with a filament current of 2.2 A and an accelerating voltage of 2.0 kV. To prepare samples for cross-sectional analysis, the film was carefully pulled apart and broken into flakes. Pitch length was measured directly from cross-sectional images taken at various points across the film. The pitch lengths of the purple-colored and green-colored regions of the film were compared to assess their roles in color formation. To measure the sizes of the chitin nanofibers, a well-diluted dispersion was deposited onto a freshly cleaved mica substrate, dried at 50 °C, and coated with a 2 nm layer of platinum.

Atomic force microscopy (AFM, Nanocute, SII Instruments, Chiba, Japan) was used to further observe the partially deacetylated chitin nanofibers and films. A silicon cantilever with a nominal spring constant of 17 N/m (SI-DF20; Hitachi High-Tech, Tokyo, Japan, 132 kHz) was employed. Isolated nanofibers were deposited on mica using the same method used for FE-SEM, and their widths were measured based on the height information in the scanned images. More than 100 nanofibers were randomly selected to calculate their height distributions. The surfaces of the films were scanned, and surface roughness was measured.

X-ray diffraction (XRD) profiles of the partially deacetylated chitin nanofiber films and the original chitin powder were analyzed using Ni-filtered CuKα radiation from an X-ray generator (Ultima IV, Rigaku Corporation, Tokyo, Japan). The diffraction data were collected at 40 kV and 40 mA within a scanning range of 5°–45°. The crystallinity index was calculated using the following equation: Crystallinity index (%) = (I_110_ − I_am_) × 100/I_110_, where I_110_ represents the maximum intensity of the 110 planes, and I_am_ is the intensity of the amorphous diffraction at 2θ = 16° [23].

The light transmittance of the chitin nanofiber cast films was recorded over a wavelength range of 200–900 nm using a UV-Vis spectrophotometer (V550; JASCO, Tokyo, Japan). The thicknesses of the films were measured using a digital micrometer (25 MDC-25MX, Mitutoyo Co., Ltd., Kanagawa, Japan). The measurement was performed at intervals of 5 mm from edge to edge.

The viscosity of the partially deacetylated chitin nanofiber dispersion was measured before and after ultrasonication using a Brookfield digital viscometer (DV-E) equipped with spindle no. LV-2 (Brookfield Engineering Laboratories, Middleboro, MA, USA) at a solid content of 1 wt%.

## 3. Results and Discussion

### 3.1. Ultrasonic Treatment of Partially Deacetylated Nanochitin

In this study, ultrasonic treatment was applied to 50% deacetylated chitin nanofibers to reduce fiber entanglement and improve the fluidity of the dispersion. This modification facilitated the self-assembly of the nanofibers, leading to the formation of structural color. AFM and FE-SEM images of the nanochitin after ultrasonic treatment are displayed in Figure 2. The ultrasonic treatment significantly reduced the size and enhanced the uniformity of the nanochitin, as illustrated in Figure 3. The nanochitin fibers exhibited similar shapes and a narrow length distribution, with the majority of fibers measuring between 100 and 150 nm and possessing an average length of 133 nm. The width distribution, estimated from AFM height measurements, was also narrow, with most fibers having a width between 3 and 5 nm and an average fiber width of 5 nm. Independent short chitin fibers were observed consistently across the field of view, indicating a successful reduction in fiber length and uniform dispersion. The ultrasonic treatment significantly reduced the viscosity of the dispersion, dropping from 14,110 mPa·s to 397 mPa·s after 8 min of treatment. The shortened fibers exhibited less resistance to flow, making the dispersion more suitable for further processing. These results demonstrated that the ultrasonic treatment effectively converted partially deacetylated nanochitin into shorter nanowhiskers. The decrease in viscosity is directly related to the reduced aspect ratio of the nanowhiskers. The X-ray diffraction profiles of the chitin powder and partially deacetylated nanochitin both before and after ultrasonication are displayed in the 5° ≤ 2*θ* ≤ 35° range (Figure 4). The peaks at 2θ = 9.46° and 19.6° correspond to the (020) and (110) planes of the chitin crystal, respectively. The peaks of the chitin nanofiber treated with ultrasonic waves had lower intensities than the original and untreated chitin nanofiber. The relative crystallinity index, which was estimated from the ratio of the crystalline peak’s intensity to the entire intensity, were found to be 88%, 87%, and 66% for chitin powder, untreated nanochitin, and treated nanochitin, respectively. This suggests that sonication reduced the crystallinity of chitin.

### 3.2. Structural Colors of Cast Films of Partially Deacetylated Chitin Nanowhiskers

A water dispersion of 1 g of partially deacetylated chitin nanowhiskers with a small aspect ratio prepared via sonication was poured into a polystyrene dish and slowly dried at 60 °C. Drying took only 2 h, and the resulting thin dried films were easily removed from the Petri dish. The films showed a characteristic color and pattern consisting mainly of green and purple colors, which were particularly clear on a black background (Figure 5a,a1,a2). Drying at 30 °C did not change the color or pattern of the films. An AFM image of one of the film’s surfaces is shown in Figure 5d. The average height difference obtained from the AFM image is 21 nm, which is considerably lower than the wavelength of visible light. This indicates minimal scattering of visible light on the film’s surface [24]. An FE-SEM image of the film’s cross section is shown in Figure 5d. The films were multilayered. The spacing between each layer was 353 ± 93 nm. In other words, the coloration of the film was found to be a structural color derived from the interference of light in the multilayer structure with spacing around the wavelength of visible light (approximately 360–830 nm). 

The distance between layers was measured in both the adjacent purple-like and green-colored regions of the film (Figure 5b,c). The thickness of the purple-like region was 3.5 μm, while that of the green-colored region was 2 μm. In the purple-like region, the average pitch length was determined to be 270 ± 30 nm, whereas in the green region, it was 330 ± 20 nm. The reflection of light from these layers was estimated using Bragg’s Law, represented as λ = n_avg_ × P, where n_avg_ is the mean refractive index (approximately 1.55 for chitin) and P is the pitch length [25]. The corresponding wavelength of reflected light falls within the range of 372–465 nm for the purple-like region and 480–542 nm for the green region. Since these wavelength ranges are within the visible spectrum, the predominant observed colors are likely to be purple-like in the first region and green in the second. The shift in color between these regions is strongly influenced by the variation in pitch length, where shorter pitch lengths reflect shorter wavelengths (closer to violet) and longer pitch lengths reflect longer wavelengths (closer to green). These findings demonstrate that the structural spacing between the layers plays a crucial role in determining the observed color of the film. The variation in pitch length across the film’s surface causes selective reflection of specific wavelengths of light, resulting in the color differences between the purple-like and green regions. This provides strong evidence that the observed interference colors are directly related to the periodic structuring of the chitin layers, further supporting the role of nanoscale organization in producing the film’s optical properties. The variation in pitch length may be influenced by capillary flow during drying, which causes uneven nanofiber deposition and potentially contributes to the observed differences.

The intensity of the structural color of the film varied with the observation angle. When the angle of observation was perpendicular to the film, the structural color disappeared. This is because the reflectance of light is negligible when the angle of incidence of the visible light is small. In contrast, the reflectance increased and the intensity of the structural color increased when the angle was greater than 45°. The patterns remained consistent in appearance when the viewing angle was changed, yet it appeared to shift toward the edge of the film at the same time as the color changed. This is because the optical path difference determines the wavelength of interfering light. In other words, the larger the angle of incidence, the smaller the optical path difference, and the shorter the wavelength of the intensifying light. The structural color of the film was not uniform, with concentric stripes developing from the center towards the edges, and the color intensity decreased towards the edges. This suggests that the film was anisotropic. The film’s thickness profile is displayed in Figure 6. The average thickness of the cast film differed significantly between the center and edges, being 5.58 ± 1.67 µm and 10.08 ± 10.04 µm, respectively. The film’s thinnest area was its eye, which gradually thickened toward the edge. As discussed below, the thinner the film, the stronger the structural color. This difference is due to the coffee ring effect [26]. This effect implies that the water droplets on a substrate evaporate from the edges, and the water in the center moves toward the edges. This causes the nanofibers in the dispersion to move away radially from the center, resulting in a gradient in the film thickness between the center and the edges.

The structural color of the cast films made from partially deacetylated nanochitin that was not sonicated was weak. This is likely due to the untreated nanochitin’s high aspect ratio and high viscosity, which prevents the formation of an ordered layered structure necessary for light interference in the visible spectrum [27]. Without proper alignment of the nanofibers, the film lacked the periodic structural organization required to reflect and refract light at specific wavelengths, leading to diminished color intensity. The lack of sonication limited the uniformity and self-assembly of the nanofibers, reducing the film’s ability to create the interference patterns responsible for strong structural colors. Therefore, the morphological control of nanochitin via sonication is necessary to develop a structural color. Even when the ultrasonic treatment time was increased to 16 min, no visible change in color was observed, indicating that 8 min of treatment was sufficient. The UV-Vis spectra of the partially deacetylated nanochitin cast films before and after sonication are shown in Figure 7. Because the ultrasonically treated film was multilayered, incident light was repeatedly reflected from each layer, producing a wavy interference spectrum [28]. In contrast, the untreated film did not exhibit a wavy spectrum, suggesting that it did not have an ordered multilayer structure with a pitch length corresponding to the visible light region. This supports the idea that the coloring mechanism is a structural color owing to the multilayer structure. The ultrasonic treatment improved the transparency of the films. For example, the transmittance of the films at a wavelength of 600 nm improved by 2.4%. This is because ultrasonic treatment shortens the fibers and improves their fluidity in water; thus, the nanochitin packs densely into the film as water evaporates. This is due to the reduced light scattering caused by the reduced voids in the film and the improved smoothness of the film surface.

### 3.3. Effect of the Weight of the Nanochitin Dispersion on the Structural Color of the Film

The weight of the partially deacetylated chitin nanowhisker dispersion was varied to study its effect on the structural color of the resulting cast film. When the weight of the dispersion was increased from 1 to 3 or 4 g, the intensity of the visual structural color decreased significantly (Figure 8). This is presumably due to changes in film thickness and multilayer pitch length. The film thicknesses were 6.18, 17.35, and 24.83 mm when the dispersion weights were 1, 3, and 4 g, respectively. Multilayer formation was observed in all the cast films, with layer spacings of 353 ± 93, 362 ± 76, and 383 ± 69 nm, respectively. Based on these figures, the number of layers increased with the film thickness. As visible light passed through numerous layers, it was repeatedly reflected, refracted, and scattered, rendering it unsuitable for structural coloration. Thus, the thin films were advantageous for structural coloration. The transmission spectra of the films were recorded, as shown in Figure 8c. All the films maintained high transparency and approximately the same transmittance. In contrast, for the wave form of the interference spectrum, a decrease in intensity and shortening of the period were observed in the 3 g film compared to the 1 g film, and no interference spectrum was observed in the 4 g film. This result supports the above discussion concerning how increasing the dispersion weight increases the number of layers, resulting in non-uniform spacing between layers and less light interference. However, when the weight of the dispersion was reduced to 0.5 g, the structural color strength of the film increased, as expected; however, it was too thin and difficult to remove from the Petri dish. Thus, the optimum dispersion weight for fabricating a film with structural color was determined to be 1 g.

### 3.4. Effect of Salt Concentration of Chitin Nanowhisker Dispersion on Structural Color of Films

To investigate the effect of the salt concentration in the dispersion on structural color, partially deacetylated chitin nanowhiskers were dialyzed in water containing different concentrations of NaCl. Subsequently, 1 g of the dispersion was dried to obtain a cast film. The appearances of these films are shown in Figure 9. The intensity and pattern of the structural color of the films observed through visual inspection were similar at NaCl concentrations of 0–10 mM. In contrast, the color intensity decreased markedly at 20 mM and nearly disappeared at 40 mM. To verify this, the transmission spectra of the films were measured (Figure 9f). The changes in the spectra were correlated with the changes in the appearance of the film. The amplitude and intensity of the waves in the interference spectrum decreased with an increasing salt concentration. Thus, the transmission rate decreased. This is related to a decrease in dispersibility owing to a decrease in the nanofiber surface charge and an increase in the ionic strength of the dispersion [29]. Poor dispersion led to aggregation and reduced nanowhisker mobility, inhibiting the formation of periodic multilayer structures. Therefore, reducing the ionic strength of water is preferable for producing films with stronger structural colors.

## 4. Conclusions

Partially deacetylated chitin nanowhiskers were used to create structurally colored films. This idea was inspired by the higher-order structure of beetle elytra. Because of the gradation of the film thickness caused by the coffee ring effect, the structural colors show a characteristic pattern of concentric stripes. The spacing between layers corresponds to the wavelength of visible light, enhancing certain visible rays through interference. To develop a structural color, nanochitin must be made more fluid in water to promote its self-assembly. In this study, this was accomplished through the simple sonication of nano-chitin. The optimum film thickness and salt concentration of the dispersion required for the appearance of a strong structural color were determined. Any color can be created freely if the spacing between the layers can be controlled. Unlike ordinary dyes, which develop color via light absorption, structural colors are weather-resistant and do not fade. Nanochitin is expected to have industrial applications as a functional dye in the future.

## Figures and Tables

**Figure 1 materials-17-05357-f001:**
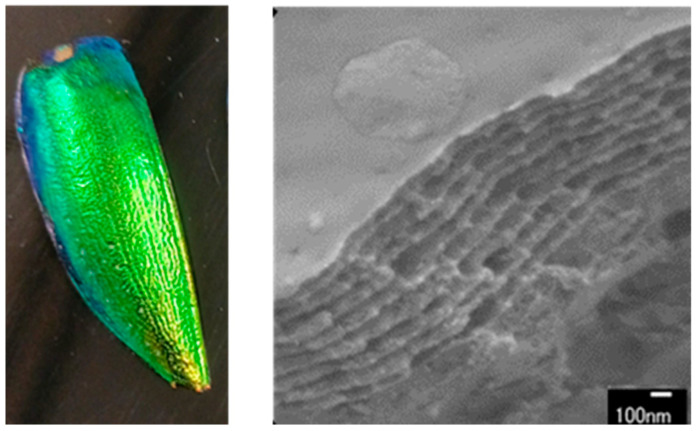
Elytron of jewel beetle with structural color and its cross section.

**Figure 2 materials-17-05357-f002:**
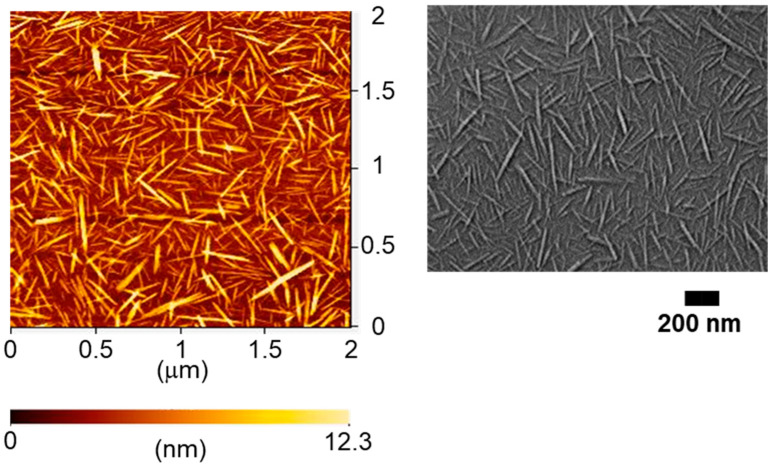
AFM and SEM images of partially deacetylated chitin nanowhiskers after ultrasonic treatment.

**Figure 3 materials-17-05357-f003:**
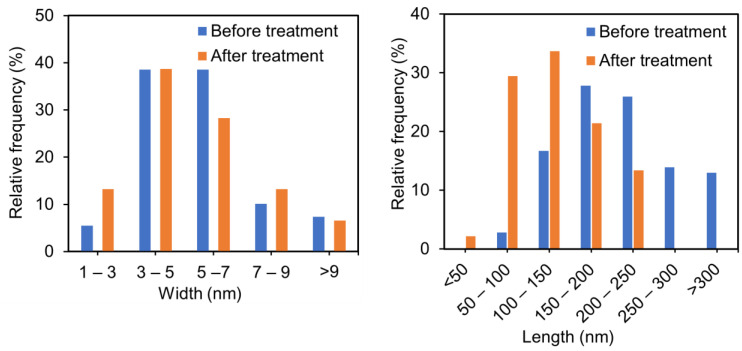
Width and length of partially deacetylated chitin nanowhiskers before and after ultrasonic treatments.

**Figure 4 materials-17-05357-f004:**
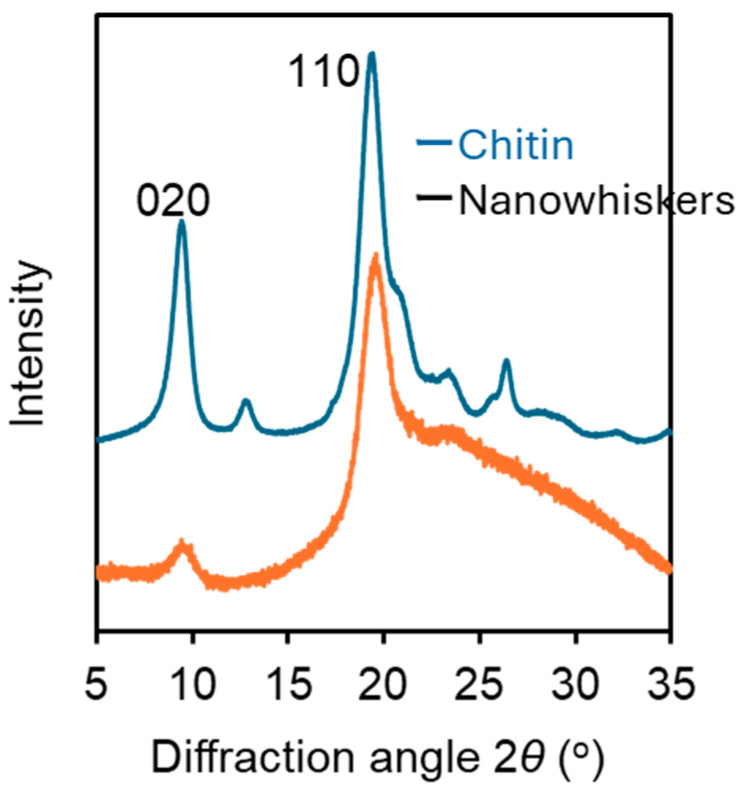
X-ray diffraction profile of chitin powder and partially deacetylated chitin nanowhiskers after ultrasonic treatment.

**Figure 5 materials-17-05357-f005:**
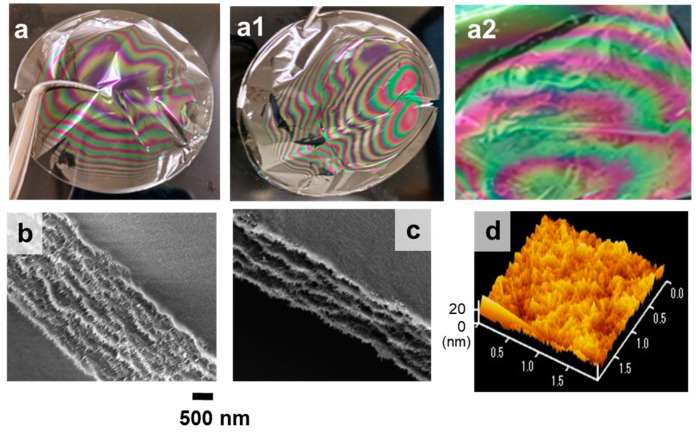
(**a**,**a1**,**a2**) Cast films of partially deacetylated chitin nanowhiskers, (**b**) SEM image of the cross-section in the purple-like region, (**c**) SEM image of the cross-section in the green-colored region, and (**d**) AFM images of the film surface.

**Figure 6 materials-17-05357-f006:**
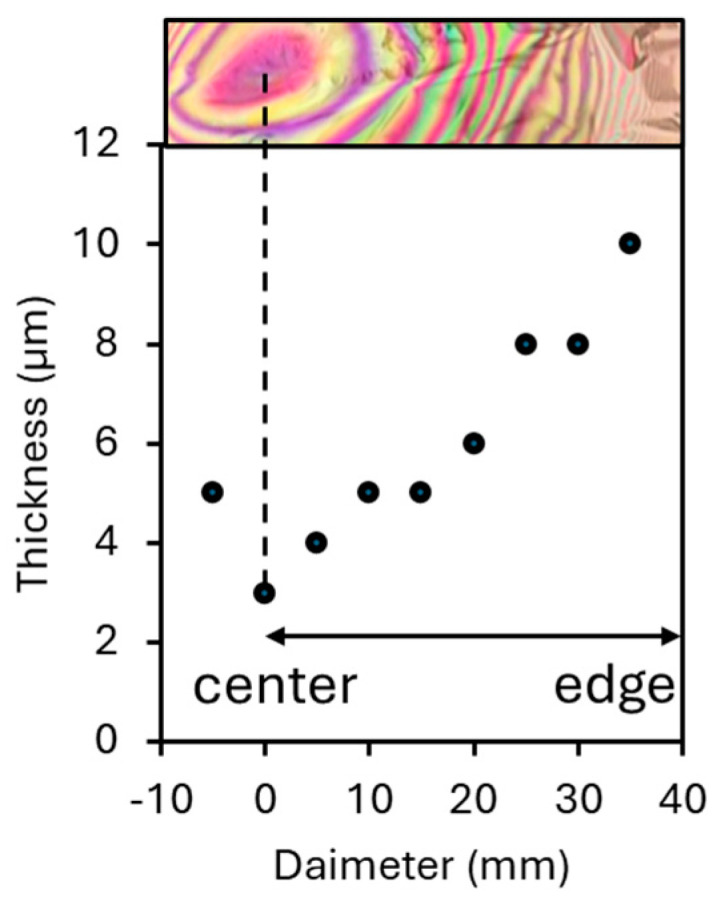
Thickness of cast film at different positions from center to edge.

**Figure 7 materials-17-05357-f007:**
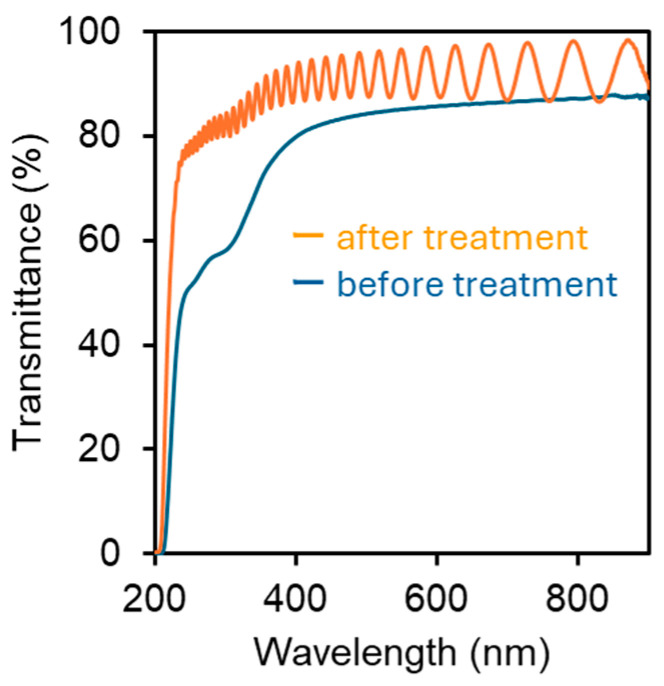
UV-Vis spectra of partially deacetylated chitin nanowhiskers with and without ultrasonic treatment.

**Figure 8 materials-17-05357-f008:**
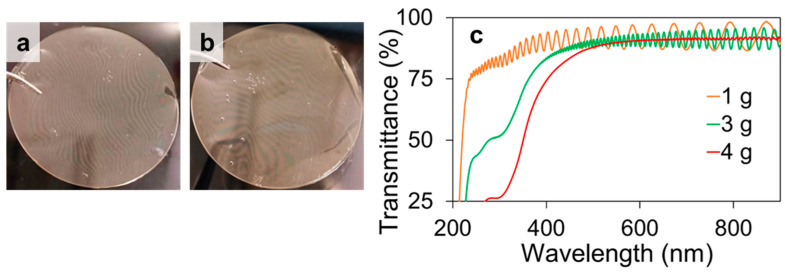
Cast films of partially deacetylated chitin nanowhiskers prepared from (**a**) 3 and (**b**) 4 g of dispersion, and (**c**) the corresponding UV-Vis spectra.

**Figure 9 materials-17-05357-f009:**
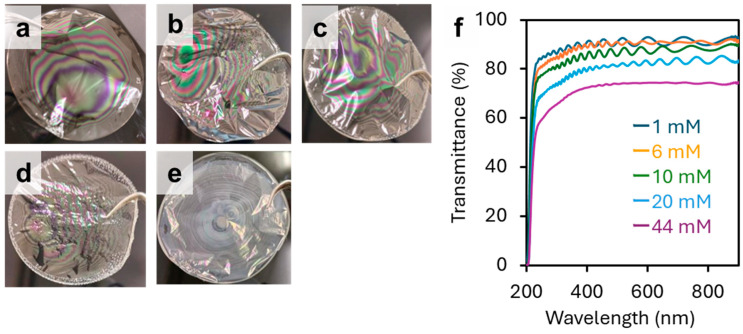
Cast films of partially deacetylated chitin nanowhiskers with different NaCl concentrations, namely, (**a**) 1, (**b**) 6, (**c**) 10, (**d**) 20, and (**e**) 44 mM, and (**f**) the corresponding UV-Vis spectra.

## Data Availability

The original contributions presented in the study are included in the article, further inquiries can be directed to the corresponding author.

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
