# Peer review of "Structural Color of Partially Deacetylated Chitin Nanowhisker Film Inspired by Jewel Beetle"

_materials, 2024, doi:10.3390/ma17215357_

Round 1
Reviewer 1 Report
Comments and Suggestions for Authors
The manuscript “Structural color of partially deacetylated chitin nanowhisker film inspired by jewel beetle” reports on the preparation of thin films of commercially available nanochitin, prepared by simple evaporation of a sonicated solution including some salt. The resulting films displayed colorful bands corresponding to variations in the thickness of the resulting film, as demonstrated by the authors, although the method of measuring film thickness at different points was not explained.
The authors claim the observed interference colors to be the result of the periodically layered structure of the film, however, they present no evidence for this claim, nor do they demonstrate that the color pattern changes with changing pitch length.
Instead, the results indicate clearly that the resulting structural colors are purely the effect of thin film interference that can be produced with any material with a refractive index different from that of the medium, a phenomenon they are familiar with, as they mention it in the introduction. The reasons for differences in the color intensity are either differences in film thickness (well demonstrated) and salt concentration, which makes the film more opaque, likely due to salt crystal formation (clearly visible on the presented figures).
Ther is no convincing evidence for:
- the observed interference colors being the result of chitin structuring
- the differences in the transmittance of light being in any way related to chitin structuring
- any kind of ordering of the deposited chitin apart from the (poorly) imaged layers
- The distance between layers of repeating chitin orientation affecting the wavelength of observed colours
The claims of the manuscript are therefore not in a any way supported by the results, and I cannot recommend its publication.
Author Response
Dear reviewer #1,
I am sending herewith the revised research article entitled “Structural color of partially deacetylated chitin nanowhisker film inspired by jewel beetle” which I should like to submit for publication in Materials.
We sincerely thank you for your time and effort in reviewing our manuscript. We appreciate your insightful comments and constructive suggestions, which have helped us improve the clarity and quality of our work. Below, we provide detailed responses to each of your comments and have made the corresponding revisions to the manuscript.
- The manuscript “Structural color of partially deacetylated chitin nanowhisker film inspired by jewel beetle” reports on the preparation of thin films of commercially available nanochitin, prepared by simple evaporation of a sonicated solution including some salt. The resulting films displayed colorful bands corresponding to variations in the thickness of the resulting film, as demonstrated by the authors, although the method of measuring film thickness at different points was not explained.
Thank you for taking the time to review our manuscript. We greatly appreciate your comments, which have helped us refine our work. In response, we have updated the manuscript to include the measurement points.
- The authors claim the observed interference colors to be the result of the periodically layered structure of the film, however, they present no evidence for this claim, nor do they demonstrate that the color pattern changes with changing pitch length.
Thank you for your insightful comment. We understand your concern regarding the relationship between the periodic structure of the film and the observed interference colors. Our current data demonstrates the presence of a periodic structure through cross-sectional FE-SEM image analysis, revealing the layered arrangement of the nanowhiskers. Additionally, interference effects observed in the UV-Vis spectroscopy measurements further support this conclusion, showing distinct peaks that correspond to the wavelengths of visible light.
The average pitch length, as measured from the FE-SEM images, was found to be 353±93 nm, which is within the visible light wavelength range, that aligns with the expected conditions for the formation of interference colors. While we did not directly vary the pitch length in this study, the consistency between the measured pitch length and the observed interference colors suggests a strong correlation between the film’s structure and its optical properties. Based on these observations and the theoretical background, we concluded that the layered structure of the film is responsible for the formation of interference colors.
We acknowledge that direct experimental data showing changes in color patterns with varying pitch length would provide further validation, and we aim to address this aspect in future studies. We hope this explanation clarifies our conclusions in the current manuscript.
- Instead, the results indicate clearly that the resulting structural colors are purely the effect of thin film interference that can be produced with any material with a refractive index different from that of the medium, a phenomenon they are familiar with, as they mention it in the introduction. The reasons for differences in the color intensity are either differences in film thickness (well demonstrated) and salt concentration, which makes the film more opaque, likely due to salt crystal formation (clearly visible on the presented figures).
Thank you for your thoughtful observations. We understand your concern regarding the role of thin film interference in our film. However, we believe that the unique contribution of the periodic arrangement of chitin nanowhiskers also influences the optical properties of the resulting colors. While thin-film interference can indeed occur with any material that has a different refractive index than its medium, our work focuses on how the self-assembly of chitin nanowhiskers into a periodic layered structure enhances and stabilizes these interference effects.
Regarding the differences in color intensity, we agree that variations in film thickness are one of the factors, as demonstrated in our results. In our study, the film thickness varied between 3 and 10 micrometers. This range of thickness likely minimizes the influence of simple thin-film interference, allowing the periodic structure of the chitin layers to have a more prominent role in the observed coloration.
We also appreciate your comment on the role of salt concentration in increasing the film’s opacity. Addition of salt plays a role in lowering the electrostatic repulsion between chitin nanocrystals, which may lead to the formation of nanochitin aggregates. This aggregation reduces the uniformity of the layered structure, thereby diminishing the interference effect and lowering the film's transmittance. This phenomenon likely contributes to the increased opacity observed in films with higher salt concentrations, as seen in the figures presented.
We hope that this response, along with the revisions made, addresses your concerns and provides a clearer understanding of the role that the chitin-based periodic structure plays in the formation of the observed structural colors.
- Ther is no convincing evidence for:
- the observed interference colors being the result of chitin structuring
- the differences in the transmittance of light being in any way related to chitin structuring
- any kind of ordering of the deposited chitin apart from the (poorly) imaged layers
- The distance between layers of repeating chitin orientation affecting the wavelength of observed colours
The claims of the manuscript are therefore not in a anyway supported by the results, and I cannot recommend its publication.
Thank you for your detailed comments. We appreciate the opportunity to address these concerns and clarify our findings. Below, we provide explanations for each of your points:
- Regarding “the observed interference colors being the result of chitin structuring.”
We understand your concern regarding the evidence for interference colors resulting from chitin structuring. Our conclusion is based on cross-sectional FE-SEM images, which suggest a periodic arrangement of nanochitin layers, with pitch length measurements falling within the visible light range. Additionally, UV-Vis spectroscopy data shows peaks that correspond to these wavelengths, indicating a relationship between the layered structure and the observed colors. While we acknowledge that more direct evidence, such as a systematic variation in layer spacing, would further strengthen this claim, we believe our current data provides a reasonable indication of the role of chitin structuring in the formation of interference colors.
- Regarding “the differences in the transmittance of light being in any way related to chitin structuring”
We appreciate your concern regarding the relationship between the light transmittance and chitin structuring. The variations in transmittance and interference observed in our study are partly influenced by the arrangement of chitin layers, as these structured layers can scatter and reflect light, impacting the film’s overall transparency and color. Additionally, changes in ionic strength alter the electrostatic interactions between chitin nanocrystals, leading to aggregation and reducing the uniformity of the layers. This led to lower transmittance and loss of color. While the role of chitin structuring may not be the sole factor influencing transmittance, it contributes to the optical properties of the film in combination with other factors, such as ionic strength and film thickness. We hope that the revisions we have made and the explanations provided address your concerns.
- Regarding “Any kind of ordering of the deposited chitin apart from the (poorly) imaged layers.”
Thank you for your observation regarding the ordering of the deposited chitin. We acknowledge that the cross-sectional FE-SEM images may not fully capture the detailed ordering of the layers. This limitation is partly due to slight damage to the layered structure during sample preparation, particularly when pulling the film apart. The stresses involved in this process can disturb the arrangement of the delicate nanochitin layers.
However, we have performed a series of imaging on multiple samples, all of which confirmed similar results. The ordering is consistent across all samples, with an average distance between layers that falls within the specified range. These observations support our conclusion that the layered structure contributes to the interference effects observed in our study.
We have revised the manuscript to clarify the potential impact of sample preparation on the imaging results.
- Regarding “the distance between layers of repeating chitin orientation affecting the wavelength of observed colors.”
We appreciate your concern regarding to the effect of distance between layers on the wavelength of observed colors. We understand that direct evidence of how the distance between layers affects the observed wavelengths would provide stronger support for our conclusions. While our current data shows that the pitch length measured from FE-SEM images falls within the visible light range, suggesting that this periodicity could contribute to interference effects. We did not perform a systematic variation of layer spacing to directly correlate it with changes in color. We acknowledge its importance and recognize that it is an area for further study.
Reviewer 2 Report
Comments and Suggestions for Authors
In the abstract section, the authors should emphasize the significance of structural coloration and its potential applications. Additionally, they should clarify the importance of maintaining low salt concentration for optimal dispersion fluidity. A brief mention of the practical applications of the resulting films would enhance the research's relevance.
The introduction provides a solid overview of chitin and its applications but would benefit from improved coherence and additional references to strengthen the background.
The authors could enhance clarity by streamlining the description of materials and methods. Specifically, they should simplify the preparation steps for the iridescent film and the characterization techniques, reducing redundancy.
Lines 127 to 129 mention that "the treatment of chitin with sodium hydroxide yields derivatives with partially deacetylated surfaces, causing 50% of the amine groups (-NH2) to be exposed on the surface." However, this statement lacks sufficient justification, as the techniques presented do not allow for the analysis of functional groups. To strengthen this section, it would be beneficial to include specific methods used for assessing the deacetylation process and the exposure of amine groups,
From lines 126 to 138, this section reads more like an introduction than results, and it should not be included in this part.
Comments on the Quality of English LanguageThe English level is clear and coherent, though some sections could be more concise.
Author Response
Dear reviewer #2,
I am sending herewith the revised research article entitled “Structural color of partially deacetylated chitin nanowhisker film inspired by jewel beetle” which I should like to submit for publication in Materials.
Thank you for taking the time to review our manuscript and for providing your insightful feedback. Your comments and suggestions have been valuable in refining our work. We have carefully considered your input and made the necessary revisions to address your concerns.
- In the abstract section, the authors should emphasize the significance of structural coloration and its potential applications. Additionally, they should clarify the importance of maintaining low salt concentration for optimal dispersion fluidity. A brief mention of the practical applications of the resulting films would enhance the research's relevance.
Thank you very much for your valuable comments. We have revised the abstract accordingly, incorporating the significance of structural coloration, the importance of maintaining low salt content, and its potential applications.
- The introduction provides a solid overview of chitin and its applications but would benefit from improved coherence and additional references to strengthen the background.
Thank you very much for your insightful comments. We have enhanced the coherence of the introduction and added the necessary references.
- The authors could enhance clarity by streamlining the description of materials and methods. Specifically, they should simplify the preparation steps for the iridescent film and the characterization techniques, reducing redundancy.
Thank you for your insightful comments on our work. We have revised the materials and method section to improve the clarity and conciseness, and to remove any redundancy.
- Lines 127 to 129 mention that "the treatment of chitin with sodium hydroxide yields derivatives with partially deacetylated surfaces, causing 50% of the amine groups (-NH2) to be exposed on the surface." However, this statement lacks sufficient justification, as the techniques presented do not allow for the analysis of functional groups. To strengthen this section, it would be beneficial to include specific methods used for assessing the deacetylation process and the exposure of amine groups,
Thank you very much for your valuable suggestions. We fully agree with your comments. The 50% partially deacetylated chitin nanofibers were provided by Marin Nanofiber Co., Ltd., and all related analyses were carried out in the company. We have removed the mentioned section, as it may have sounded more appropriate for the introduction.
- From lines 126 to 138, this section reads more like an introduction than results, and it should not be included in this part.
Reviewer 3 Report
Comments and Suggestions for Authors
The papers deals with the preparation of structured coloured films formed by nanowhiskers of partially deacetylated chitin.
The abstract must be improve. It is not conclusive.
It is not clear why ultrasonication can form nanowiskers from the partial deacetylated chitin.
Why the proposed material is bioinspired from the coloured jewel beetle?
Which are the colours diplayed by the nanowiskers-based film?
Line 92 Which ultrasonic treatment refers to?
Author Response
Dear reviewer #3,
I am sending herewith the revised research article entitled “Structural color of partially deacetylated chitin nanowhisker film inspired by jewel beetle” which I should like to submit for publication in Materials.
Thank you for the time you invested in reviewing our manuscript, as well as for your thoughtful questions and suggestions. Your feedback has been instrumental in clarifying key aspects of our work and improving its overall presentation. We have carefully addressed each of your questions and incorporated your suggestions into the revisions.
- The abstract must be improved. It is not conclusive.
Thank you for your valuable comments. We have revised the abstract and incorporated additional information to enhance its clarity and content.
- It is not clear why ultrasonication can form nanowiskers from the partial deacetylated chitin.
Thank you for your time and for your insightful questions, which have helped clarify our work. We have updated the manuscript to improve its clarity.
Partial deacetylation introduces amine functional groups (-NH2), which become protonated in acidic conditions (-NH3+). This protonation enhances the stability of the dispersion through electrostatic repulsion. The high-energy mechanical forces generated by ultrasonication further break down the chitin structures, resulting in the production of thinner, shorter, and more uniform nanowhiskers.
- Why the proposed material is bioinspired from the colored jewel beetle?
Thank you for your valuable comments. We have updated the manuscript accordingly.
The reason for the bioinspiration of the proposed material is that the beetle's vivid iridescent colors result from its unique micro- and nanoscale chitin structures, which interact with light to produce structural coloration. By replicating these natural structures, our material aims to achieve similar optical effects, using the arrangement of chitin nanostructures to create vibrant, durable colors without synthetic dyes. This bioinspired approach enables the development of environmentally friendly, non-toxic materials. Additionally, the material's responsiveness to environmental changes, biocompatibility, and non-toxicity makes it well-suited for applications in food sensors and cosmetics.
- Which are the colors displayed by the nanowiskers-based film?
Thank you very much for taking the time to review our manuscript. Your questions and suggestions have helped us improve the clarity of our work. We have addressed your concerns as follows:
The nanowhisker-based film displays a range of vibrant colors, primarily green and purple, depending on the thickness of the film and the arrangement of the nanostructures. These colors result from the interference of light within the multilayered structure of the film. Variations in film thickness and the periodic spacing between layers can shift the reflected wavelengths, thereby producing different colors. This optical phenomenon is consistent with structural coloration, similar to that observed in nature, such as in beetle exoskeletons.
- Line 92 Which ultrasonic treatment refers to?
Thank you for your valuable comments. In response, we have revised the materials and methods section to enhance clarity. We have specifically addressed the details of the ultrasonic treatment, providing a clearer and more precise description in the updated manuscript.
Round 2
Reviewer 1 Report
Comments and Suggestions for Authors
I do not find that the changes made to the manuscript addressed any of its major issues.
Author Response
Dear Reviewer #1,
Thank you very much for your thoughtful comments and for the opportunity to revise our manuscript once again. We have carefully considered your feedback and made further revisions to clarify our findings. We hope that this updated version addresses your concerns. We appreciate your time and effort in helping us improve the quality of our work.
- The manuscript “Structural color of partially deacetylated chitin nanowhisker film inspired by jewel beetle” reports on the preparation of thin films of commercially available nanochitin, prepared by simple evaporation of a sonicated solution including some salt. The resulting films displayed colorful bands corresponding to variations in the thickness of the resulting film, as demonstrated by the authors, although the method of measuring film thickness at different points was not explained.
Thank you very much for taking the time to review our manuscript and response. Regarding the thickness measurement, we measured the film from one edge to the other at 5 mm intervals. The manuscript has been updated to reflect this revision.
- The authors claim the observed interference colors to be the result of the periodically layered structure of the film, however, they present no evidence for this claim, nor do they demonstrate that the color pattern changes with changing pitch length.
Thank you very much for your comments and opportunity to revise our document for the second time. In the revised manuscript we have taken samples from adjacent purple like and green color part of the film and measured the pitch length. The reflection of light was also estimated using Bragg’s Law. The result indicates that the structural spacing in this region corresponds to wavelengths of light responsible for producing purple-like and green colors. This supports the idea layered structure of the film is responsible for the formation of interference colors.
- Instead, the results indicate clearly that the resulting structural colors are purely the effect of thin film interference that can be produced with any material with a refractive index different from that of the medium, a phenomenon they are familiar with, as they mention it in the introduction. The reasons for differences in the color intensity are either differences in film thickness (well demonstrated) and salt concentration, which makes the film more opaque, likely due to salt crystal formation (clearly visible on the presented figures).
Thank you for your comments and for the opportunity to revise our manuscript again. Our findings suggest that variations in color are influenced by the self-assembled nanostructures within the film. We measured the pitch length variations between adjacent purple and green regions of the film to assess their relationship with the observed colors. The results show that structural spacing directly corresponds to the observed purple and green colors. This data has been included in the updated manuscript.
Additionally, the transmittance data comparison between the ultrasonicated and untreated films, prepared under the same conditions, clearly demonstrates that the treated film exhibits sinusoidal oscillations, while the untreated film does not. This indicates that ultrasonication enhances the ordering of the layers, improving their interaction with light. In contrast, the untreated film, due to its less ordered structure, does not produce these sinusoidal oscillations or show similar color effects. This further supports our claims about the influence of nanostructural ordering on the optical properties of the film.
- Ther is no convincing evidence for:
- the observed interference colors being the result of chitin structuring
- the differences in the transmittance of light being in any way related to chitin structuring
- any kind of ordering of the deposited chitin apart from the (poorly) imaged layers
- The distance between layers of repeating chitin orientation affecting the wavelength of observed colors
Thank you again for reviewing our manuscript and response. We appreciate the opportunity to clarify our findings regarding chitin structuring and its influence on the observed interference colors. We understand the need for more convincing evidence, and we have addressed each of your concerns as follows:
Interference colors as a result of chitin structuring: In the revised manuscript, we have included pitch length measurements from different parts of the film and correlated these with the observed colors. The variation in pitch length across different regions of the film aligns with the changes in color, supporting the hypothesis that the nanostructural arrangement influences the interference pattern.
Differences in transmittance of light related to chitin structuring: As indicated in the manuscript, films were prepared using both ultrasonicated and untreated partially deacetylated chitin under identical conditions. The treated film exhibited color, while the untreated one did not. We compared the transmittance data for both samples, as shown in the graph provided. The untreated film shows lacks significant oscillations, suggesting minimal periodicity or structural ordering. In contrast, after ultrasonication treatment, the appearance of distinct oscillations strongly indicate changes in the film’s structural organization. This enhanced periodicity likely results from a more ordered arrangement of the chitin layers, influencing how light transmits and reflects through the film. Therefore, the differences in transmittance directly correspond to the level of structural ordering within the chitin film.
Ordering of the deposited chitin layers: We acknowledge that the imaging of the chitin layers in the previous version of the manuscript might not fully demonstrate the level of ordering. We have now improved the imaging quality that better captures the layered structure of the deposited chitin. These revised images support our conclusion that there is some level of ordering influencing the optical properties.
Distance between layers affecting the wavelength of observed colors: In the revised manuscript, the pitch length measurements performed on purple and green colored part of the film indicate that variations in the distance between layers are associated with differences in the wavelength of reflected light, contributing to the observed colors. We have revised the manuscript to clarify this relationship and provide stronger support for this claim.
Reviewer 3 Report
Comments and Suggestions for Authors
The manuscript is suitable for publication
Author Response
Thank you very much for your thoughtful comments and for the opportunity to revise our manuscript. We are pleased that you decided our manuscript is suitable for publication. We appreciate your time and effort in helping us improve the quality of our work.
Round 3
Reviewer 1 Report
Comments and Suggestions for Authors
The added text and measurements make sense, but they are not convincing. Firstly, they are not supported by images that would show changes in pitch length or the correspondance of these changes to different observed colors; secondly, such a correspondance would neccessitate that the pitch lenght changes periodically with film thickness, as do the imaged colors, which is not demonstrated. I therefore do not see an improvement in the manuscript.
Author Response
Dear Reviewer #1
Thank you very much for taking the time to review our responses and evaluate the updated manuscript. We sincerely appreciate all of your constructive comments and suggestions, which have greatly improved the quality of the manuscript. We are honored to have received your valuable feedback, and we believe the manuscript is now significantly enriched.
The added text and measurements make sense, but they are not convincing. Firstly, they are not supported by images that would show changes in pitch length or the correspondence of these changes to different observed colors; secondly, such a correspondence would necessitate that the pitch length changes periodically with film thickness, as do the imaged colors, which is not demonstrated. I therefore do not see an improvement in the manuscript.
Thank you very much for your comments and suggestions. We have addressed your concern and updated the manuscript accordingly. Regarding the variation in pitch length across different regions of the film and its effect on color, we measured the thickness of the samples and reported the corresponding pitch lengths for the adjacent purple-like and green-colored regions. Cross-sectional images of both samples have been included in the updated manuscript to illustrate thickness variation alongside pitch length. We have also modified the materials and methods section to reflect these changes.